# Genetic Diversity of Selected High-Risk HPV Types Prevalent in Africa and Not Covered by Current Vaccines: A Pooled Sequence Data Analysis

**DOI:** 10.3390/ijms262211056

**Published:** 2025-11-15

**Authors:** Babalwa Nyide, Miranda Thomas, Lawrence Banks, Pamela P. Mkhize, Nontokozo D. Matume

**Affiliations:** 1Discipline of Biochemistry, School of Agriculture and Science, University of KwaZulu-Natal, Pietermaritzburg 3209, South Africamkhizep6@ukzn.ac.za (P.P.M.); 2Tumour Virology, International Centre for Genetic Engineering and Biotechnology (I.C.G.E.B.), 34149 Trieste, Italy; miranda.thomas@icgeb.org (M.T.); lawrence.banks@icgeb.org (L.B.); 3Centre for the AIDS Programme of Research in South Africa (CAPRISA), School of Medicine, University of KwaZulu-Natal, Durban 4013, South Africa; 4Discipline of Genetics, School of Agriculture and Science, University of KwaZulu-Natal, Pietermaritzburg 3209, South Africa; 5SAMRC-UNIVEN Antimicrobial Resistance and Global Health Research Unit, HIV/AIDS & Global Health Research Programme, University of Venda, Thohoyandou 0950, South Africa

**Keywords:** high-risk human papillomavirus (HR-HPV), genetic diversity, oncoproteins, vaccines, phylogenetics, Africa

## Abstract

High-risk human papillomavirus (HR-HPV) types exhibit an uneven global distribution, with types 35, 51, 56, and 59 being more prevalent in Africa yet not covered by current L1-based vaccines. The genetic diversity of HR-HPV oncoproteins in Africa remains poorly characterized, despite their potential as alternative vaccine targets. This study investigates the genetic diversity of HR-HPV types 16, 18, 35, 51, 56, and 59 to inform vaccine development. We analyzed 14,332 sequences from the NCBI Virus database and 222 HPV reference sequences from the Papillomavirus Episteme (PaVE) database using phylogenetic analysis and variant identification. HPV16 and HPV35 exhibited close evolutionary relatedness, which may indicate shared traits relevant to vaccine design, although functional implications remain to be experimentally validated. A key finding of the study was the discovery of novel non-synonymous mutations, including E148K in HPV35 E6, S63C in HPV16 E7 and S495F in HPV18 L1, as well as known oncogenic variants such as L83V (E6) and N29S (E7) in HPV16. These findings highlight significant intra- and inter-type diversity among African HR-HPVs. This study provides new insights into the genetic diversity and evolutionary relationships of underrepresented HR-HPV types. The findings underscore the need for continued genomic surveillance and support efforts to develop region-specific vaccines that include HPV35, 51, 56, and 59 to address gaps in current vaccine coverage and help reduce the burden of HPV-related cancers in Africa.

## 1. Introduction

Cervical cancer remains a major global health burden, with approximately 702,000 new cases and 370,000 deaths estimated for 2025 [1]. More than 99% of cervical cancer cases have been linked to high-risk human papillomaviruses (HR-HPVs), which are primarily transmitted through sexual contact [2,3]. Although the immune system clears around 90% of HPV infections within 6–24 months, approximately 10% of cases progress to chronic infections [4,5]. These infections lead to oncogenesis and various cancers, including cervical, anogenital, and head-and-neck cancers [6,7,8,9].

In 2018, the World Health Organization (WHO) pledged to eliminate cervical cancer as a public health problem by 2030. The global elimination strategy is built on three pillars: vaccinating 90% of girls against HPV before age 15, screening 70% of women aged 35–45 for cervical cancer, and ensuring treatment for 90% of women diagnosed with the disease [10].

Because the success of these interventions hinges on targeting the most relevant HPV types, it is critical to understand HPV risk classification. HPVs are categorized as either high-risk or low-risk based on their oncogenic potential. Phylogenetic studies have shown that HR-HPV types, including 16, 18, 31, 33, 35, 39, 45, 51, 52, 56, 58, 59, 68, and 73, belong to the closely related *Alpha-papillomavirus* genus [11]. Among these, HPV types 16 and 18 are the most prevalent worldwide, accounting for approximately 70% of all HPV-related cancer cases globally [12]. The distribution of other HR-HPV types varies across geographic regions [13,14]. Notably, epidemiological data indicate that HR-HPV types 35, 51, 56, and 59, while common globally, are disproportionately more prevalent in Africa compared to other regions [12,15].

Currently, there is no cure for HPV infections. However, six prophylactic vaccines based on the L1 capsid protein are available [16]. The first three globally approved vaccines are Gardasil^®^ (HPV6/11/16/18), Cervarix^®^ (HPV16/18), and Gardasil 9^®^ (HPV6/11/16/18/31/33/45/52/58). In 2022, two additional bivalent vaccines, Cecolin^®^ and Walrinvax^®^, both targeting HPV types 16 and 18, were approved in China [17], and a quadrivalent vaccine, Cervavac^®^ (HPV6/11/16/18), was approved in India [16,18]. All three new vaccines received WHO prequalification in 2024 [19].

Despite widespread global use, concerns remain about vaccine efficacy and coverage in diverse populations. This issue is particularly relevant for African populations, which are among the most genetically diverse in the world [20]. Of the ten most prevalent HR-HPV types among African women, four types, HPV35, 51, 56, and 59, are not covered by current vaccines [12]. This gap underscores the need for region-specific vaccines to reduce the burden of HPV-related cancers in Africa.

Advances in vaccine technology are creating opportunities to explore new candidates and strategies for HPV prevention [21]. Current research and clinical trials are investigating vaccines that target the L2 minor capsid protein and the Early (E) proteins [22]. Among these, the E5, E6, and E7 oncoproteins are of particular interest, as they play critical roles in initiating tumorigenesis and maintaining the malignant state of infected epithelial cells [23,24]. However, the diversity of these proteins may present challenges for vaccine design. Understanding the genetic variations in HR-HPVs is therefore crucial for developing vaccines with broader and more effective coverage. Although various isolated studies have explored HR-HPV genetic diversity worldwide [25,26,27,28,29,30,31,32], a comprehensive, systematic analysis of the genetic variability of multiple HR-HPV types, particularly those highly prevalent in Africa, is still lacking.

To address this gap, the present study analyzed pooled sequence data of six HR-HPV types (16, 18, 35, 51, 56, and 59) to characterize the genetic diversity of the E5, E6, and E7 oncoproteins, as well as the L1 major capsid protein. By comparing African and global sequence data, we aimed to identify Africa-specific variants and assess their potential impact on protein function. These insights are critical for guiding the design of next-generation, region-specific HPV vaccines and improving global vaccine strategies.

## 2. Results

### 2.1. Sequence Dataset Composition

A comprehensive search of the NCBI Virus Database yielded a total of 13,860 verified DNA sequences of HR-HPV types 16, 18, 35, 51, 56, and 59 from global and African sources (Figure 1). Of these, 1063 were near-complete (≥80%) genomes, while 12,797 were partial genomes or gene sequences. Globally, HPV16 and HPV18 had the highest representation, with 10,733 and 1063 sequences, respectively, followed closely by HPV35 with 1045 sequences. Within the African dataset, the most frequently represented type was HPV16 (432 sequences), followed by HPV35 (263 sequences) and HPV18 (60 sequences) (Figure 1). In contrast, HPV51, HPV56, and HPV59 were markedly underrepresented in both the global and African datasets. Notably, Oceania contributed sequences exclusively for HPV35. The limited number of sequences for HPV51, HPV56, and HPV59 significantly constrains the ability to make robust comparisons between these types and the more well-represented HR-HPVs.

Sequence datasets from Africa were further processed to identify and extract full-length gene sequences encoding the E5, E6, E7, and L1 proteins using their open reading frames (from start to stop codons) (Table 1). Gene sizes for each protein corresponded closely with those of the respective reference genomes, although some variations were observed across HPV types. Notably, gene sizes differed between the HR-HPV types, consistent with the annotated reference sequences. The E5 gene ranged from 222 bp to 252 bp (encoding 73–83 amino acids) in HPV types 16, 18, 35, and 59, but was not detected in HPV types 51 and 56, in agreement with their reference genome annotations (PaVE IDs: HPV51REF and HPV56REF, respectively). This analysis included a total of 220 full-length E5 gene sequences from ten African countries. The E6 gene ranged from 450 bp to 483 bp (149–160 amino acids), with distinct sizes observed for each HPV type. A total of 242 full-length E6 sequences were analyzed from twelve African countries. The E7 gene ranged from 297 bp to 324 bp (98–107 amino acids), with 251 full-length sequences included from eleven African countries. The L1 gene ranged from 1500 bp to 1527 bp (499–508 amino acids), with 249 full-length sequences obtained from ten African countries. In total, 962 full-length gene sequences were analyzed, distributed across HR-HPV types as follows: HPV16 (48 sequences, 5.0%), HPV18 (41 sequences, 4.3%), HPV35 (763 sequences, 79.3%), HPV51 (37 sequences, 3.8%), HPV56 (33 sequences, 3.4%), and HPV59 (40 sequences, 4.2%).

### 2.2. Phylogenetic Analysis

A total of 222 HPV reference sequences were retrieved and used for the analysis. Figure 2 illustrates the phylogenetic relationships of these sequences, including the HR-HPV consensus sequences of interest (16, 18, 35, 51, 56, and 59). The results show that the HR-HPVs of interest (highlighted in green) are closely related, clustering together within different branches of the same monophyletic clade (outlined in green), with HPV16 and HPV35 forming a distinct sub-cluster.

We compared the genetic diversity of HR-HPV consensus sequences from different regions of the world, as shown in Figure 3 (further detail is presented in Appendix A). Based on the cladograms of regional consensus sequences, HPV16 and HPV51 populations from Asia and Africa were found to be closely related. Similarly, the African HPV56 and HPV59 populations showed a close relationship with their counterparts in South America.

### 2.3. Intra-Africa Diversity

The dataset from Africa was strongly biased, with sequences for HPV types 16, 18, 51, 56, and 59 originating only from South Africa or Togo. In contrast, HPV35 sequences exhibited greater diversity, representing a total of ten countries (Figure 4 and Appendix A). Most branches on the phylogenetic trees of HR-HPVs 16, 18, 35, 56, and 59 were strongly supported (≥70%), whereas only three branches for HPV51 could be confidently resolved (Figure 4). For HPV35, a closer relationship was observed among the South African, Zimbabwean, and Togolese populations, while Moroccan and Nigerian populations clustered together. Figure 4 illustrates the phylogenetic analysis of HPV35 consensus sequences by country, with a more detailed analysis of individual isolates presented in Appendix A.

### 2.4. Oncoprotein/L1 Variant Analysis

To assess the diversity of HR-HPV oncoproteins in Africa, full-length E5, E6, E7, and L1 genes were extracted from sequences obtained from the NCBI database. Table 2, Table 3, Table 4 and Table 5 summarize variant counts, while Appendix A provide detailed non-synonymous variations. Overall, the translated sequences were highly similar to their respective reference sequences. Several polymorphisms occurred at 100% frequency, including I44L in HPV16 E5, H78Y in HPV16 E6, and T266A in HPV16 L1. In HPV16 E5, I65 was mutated in all sequences, with I65V in 10 out of 11 sequences and I65L in one sequence.

For E7, 21 non-synonymous variants were identified, and L1 proteins exhibited 55 variants across all HR-HPVs. Three novel variants were detected in African populations: E148K in HPV35 E6, S63C in HPV16 E7 and S495F in HPV18 L1. Variants present at <1% frequency were reported descriptively and excluded from statistical testing. Two-tailed Fisher’s exact tests were applied only to variants with interpretable frequencies to determine whether the distribution of HPV protein variants differed significantly across geographic regions, enabling identification of common versus rare variants and the detection of regional clustering. As these analyses were exploratory, no multiple-testing correction was applied, and findings should therefore be interpreted with caution. Several variants displayed notable geographic patterns. In HPV35 E6, W78R clustered in Algeria, Rwanda, and Togo, whereas H98Y was rare but disproportionately present in Mali. In HPV35 E7, E63K was distributed across Algeria, Guinea, Nigeria, Rwanda, and Togo. In HPV35 L1, S348T occurred in eight countries, including Algeria, Guinea, Kenya, Mali, Morocco, Nigeria, Rwanda, and South Africa, while S349T was strongly represented in Rwanda compared to other regions.

This analysis highlights both common and rare variants, identifies novel mutations, and reveals geographic clustering patterns that may inform regional vaccine design and enhance HR-HPV surveillance in Africa.

## 3. Discussion

This study provides a comprehensive analysis of the genetic diversity of six high-risk HPV types (16, 18, 35, 51, 56, and 59), with a focus on African populations compared to global sequences. By analyzing full-length genomes and key viral proteins, E5, E6, E7, and L1, our work identifies Africa-specific variations, including three novel non-synonymous mutations, E148K in HPV35 E6, S63C in HPV16 E7 and S495F in HPV18 L1, which have not been previously reported. While these findings highlight important intra- and inter-type diversity, it is important to recognize that the data are limited by the low number of available sequences for several HR-HPV types and the snapshot nature of database retrieval. Nevertheless, this study provides one of the first consolidated views of HR-HPV protein variation in Africa, establishing a baseline for future research and informing considerations for next-generation, region-specific HPV vaccines.

Previous research has largely focused on HPV16 and HPV18 due to their global prevalence, oncogenic potential, and inclusion in current vaccines [33]. Our findings suggest that HPV35, which is disproportionately prevalent in Africa, deserves greater attention. Phylogenetic analyses revealed a close evolutionary relationship between HPV35 and HPV16. While this may indicate shared biological traits, any implications for vaccine protection are speculative and require experimental validation. Similarly, evolutionary analyses indicated close relationships between HPV18 and HPV59, and between HPV51 and HPV56. Unlike prior studies that relied on concatenated amino acid sequences of selected proteins [11], our analysis of 222 complete HPV reference genomes offers a more comprehensive perspective on these relationships.

The L1 gene, highly conserved across HPV types, remains the primary target for current vaccines [17,34]. The observed L1 variants in African HPV16, 35, and 59 sequences could potentially influence antigenicity, but further functional studies are needed to determine any impact on vaccine efficacy. Notable variations such as T266A and S282P in HPV16 occur within immunogenic regions, consistent with previous findings in South African isolates that suggest these variants affect T- and B-cell epitopes [35]. Several L1 variants identified in Africa have been reported in other regions, whereas S495F in HPV18 appears novel. Variations in HPV35 and other HR-HPVs further highlight the importance of continued surveillance to inform vaccine design.

All L1 variations in HPV16 observed in Africa have also been reported at varying frequencies in Europe, Asia, the Middle East, and the Americas [35,36,37]. Similarly, L3M has been detected previously in America and Asia, while T88N, Q273P, and V323I from HPV18 have been reported in Europe [35,38]. Ahmed et al. also reported two variants (T389S and L475F) that were not observed in the present study. However, S495F in HPV18 represents a previously unreported variation in a South African isolate (OP971042.1). Additionally, S348T in HPV35 has been reported in Brazil [31], while the HPV51 variations, while V264G and G265S in HPV51 were observed in Southwest China [39].

In contrast, the E5 protein displayed considerable variability, which may limit its potential as a reliable vaccine target. Literature indicates that E5 is expressed exclusively by Alpha HPVs [40,41]. We also observed the absence of the E5 open reading frame (ORF) in HPV51 and HPV56, which has not been previously reported, suggesting that E5 may play a non-essential role in HPV-related oncogenesis and may be a poor candidate for therapeutic vaccines.

Variations in oncoproteins may influence HPV oncogenicity. Some, such as L83V in HPV16 E6 and N29S in E7, are associated with precancerous or cancerous phenotypes [42]. Combinations of multiple variations, including Q14H, H78Y, and L83V in HPV16 E6, may enhance viral immortalization and transformation of cells compared to wildtype oncoproteins [43]. These findings highlight the importance of investigating HR-HPV variations in Africa and their potential impact on vaccine-targeted antigenicity.

Comparison with the literature shows that several E6 and E7 variants identified in this study have been previously reported in other regions. For example, Q14D and H78Y in HPV16 were found in isolates from Congolese cervical cancer patients [44], while L83V in HPV16 has been reported in Europe, South/Central America, and East Asia [43]. In HPV35, I73V and W78R have been described in Brazil, Germany, and China [31,45,46,47], whereas W78N in CAR was not observed in our study [48]. E148K in HPV35 and S495F in HPV18 are novel to African isolates. Other previously described variants include S100L in HPV51 and S14R and K54N in HPV56 [46].

E7 variants in HPV16, such as N29S, have been reported in Korea and China [43], while E7Q in Chad has also been identified in CAR [48]. S63C in HPV16 from South Africa is rare and has not been widely reported, whereas other variants at this position (S63F and S63P) occur in Romania and China [43,49]. E63K in HPV35 from Africa has recently been reported in China [47]. A recent study of South African and Mozambican HPV35 isolates found SNPs in E6 and E7, though they did not result in protein variation [50].

While short-term vaccine efficacy studies in Africa are encouraging, long-term follow-up is critical due to the 3–20 year latency of HPV-related cancers [51,52,53]. The increasing prevalence of HPV35 and limited data on HPV51, HPV56, and HPV59 highlight the need for region-specific vaccine strategies. Enhanced genomic surveillance could support Africa-centric vaccine development tailored to circulating HPV variants.

This study lays a foundation for future research into Africa-specific HPV vaccines but is subject to several limitations. Sampling bias due to skewed GenBank data results in under-representation of certain regions and HR-HPV types, particularly 51, 56, and 59, limiting robust conclusions. Clinical metadata, including lesion grade, patient age, and temporal data, were not included, restricting interpretation of some findings. Additionally, the limited time window of data extraction introduces snapshot bias.

## 4. Methods and Materials

### 4.1. Database Mining and Nucleotide Sequence Retrieval

DNA sequences of HR-HPV types 16, 18, 35, 51, 56, and 59 from both global and African sources were searched and retrieved from the National Center for Biotechnology Information (NCBI) Virus Sequence Database [54]. The search was conducted using the “Virus/Taxonomy” filter with the following taxonomic identifiers: HPV16 (*Human papillomavirus 16*, taxid:333760), HPV18 (*Human papillomavirus 18*, taxid:333761), HPV35 (*Human papillomavirus 35*, taxid:10587), HPV51 (*Human papillomavirus 51*, taxid:10595), HPV56 (*Human papillomavirus 56*, taxid:10596), and HPV59 (*Human papillomavirus 59*, taxid:37115). Additionally, the search results were filtered by geographic origin using the “Geographic Region” filter to separate African sequences from those collected globally (Figure 5). All retrieved sequences were saved as separate FASTA files between August and October 2024. Complete reference genomes for HR-HPV types 16, 18, 35, 51, 56, and 59 were retrieved from the Papillomavirus Episteme (PaVE) database [55]. Microsoft Excel Professional Plus 2019 (Redmond, WA, USA) was used to check for duplicate entries. Duplicate entries, including identical accession numbers originating from the same subject or laboratory, were removed prior to downstream analyses. Geographic metadata for each sequence were validated through a two-step process: first, each location was independently verified by two reviewers; second, locations were cross-checked against the accompanying metadata to ensure consistency and accuracy. Only sequences with validated metadata were included in subsequent analyses.

### 4.2. Sequence Data Sorting and Annotation

The FASTA files containing the nucleotide sequences were imported into Geneious Prime^®^ (version 2024.0.7, Dotmatics, Boston, MA, USA), a bioinformatics software used for sequence data analysis. Each sequence was manually renamed to include its geographic origin and accession number. The following identifier codes were used to represent each world region: Afr (Africa), Asia (Asia), Eur (Europe), NAm (North America), Oce (Oceania), and SAm (South America). Specific codes were also assigned to represent the African countries from which the sequences originated: Alg (Algeria), Cong (Republic of the Congo), CAR (Central African Republic), Chad (Chad), DRC (Democratic Republic of the Congo), Egy (Egypt), Gab (Gabon), Gamb (The Gambia), Gha (Ghana), Guin (Guinea), Ken (Kenya), Mali (Mali), Maus (Mauritius), Moro (Morocco), Ngra (Nigeria), RSA (Republic of South Africa), Rwa (Rwanda), Tan (Tanzania), Togo (Togo), Tuni (Tunisia), Uga (Uganda), and Zim (Zimbabwe). Any sequence classified as “unverified” in the NCBI database was discarded. Reference genomes were manually annotated in Geneious Prime^®^, using the information provided in the GenBank files and the locus viewer of the respective reference genomes on the PaVE database [55].

### 4.3. Nucleotide Sequence Alignments and Phylogenetic Analysis

To assess genetic diversity and infer phylogenetic relationships among HR-HPV types, multiple sequence alignments were performed using MAFFT v7.525 with the FFT-NS-2 algorithm and default parameters (gap opening penalty = 1.53; gap extension penalty = 0.00) in Ubuntu (v24.04.2 LTS, Canonical, London, UK). The alignments included all verified near-complete genomes (≥80% reference sequence coverage) for each HPV type, along with their respective reference genomes. Only sequences with ≤15% ambiguous bases (N) were included, and insertions and deletions were treated as gaps in the alignments. Both coding regions and non-coding regions, including the long control region (LCR), were analyzed without masking hypervariable regions, in line with previously published literature [56]. Multiple sequence alignments were visualized and residues corresponding to gaps in the reference sequence were trimmed using AliView (v1.30, Uppsala University, Uppsala, Sweden) to reduce poorly aligned regions, ensuring consistent treatment of gaps across all sequences. Maximum likelihood phylogenetic trees were constructed using IQ-TREE v2.0.7 (multicore version) with ModelFinder for model selection and SH-aLRT/UFboot tests, each run with 1000 replicates [57]. The resulting trees were visualized and annotated using the Interactive Tree of Life (iTOL) v7 online tool [58].

### 4.4. Variation Calling and Amino Acid Translation

All verified sequences retrieved from the NCBI Virus Database for each HPV type were aligned to their respective annotated reference sequences. The genes of interest (L1, E5, E6, and E7) were identified and extracted using the “Extract” tool in Geneious^®^ Prime. Variant calling was performed using the “Find Variation/SNPs” function under the “Annotate and Predict” menu in Geneious^®^ Prime. The following data were recorded for each identified variant: Type of variant, Frequency, and Origin sequence. Incomplete gene sequences lacking identifiable start or stop codons were excluded. For variant calling, only gene sequences with 100% reference sequence coverage were included with a minimum length corresponding to the gene in the corresponding reference sequence since each HR-HPV type has a different size for the same gene (shown in Table 1). The nucleotide sequences were subsequently translated into amino acid sequences to classify mutations as synonymous or non-synonymous.

### 4.5. Statistical Analysis

Differences in the geographical distribution of variants were assessed using two-tailed Fisher’s exact tests in GraphPad Prism (v10.6.1, GraphPad Software, San Diego, CA, USA). Variants occurring at <1% frequency were reported descriptively and excluded from statistical testing to avoid over-interpretation. Statistically significant associations were indicated by asterisks (*p* < 0.05 to *p* < 0.0001), highlighting variants disproportionately represented in specific regions. No multiple-testing correction was applied, as the analyses were primarily descriptive and exploratory.

## 5. Conclusions

Our findings highlight the genetic diversity of clinically significant HR-HPV types in Africa, particularly HPV16, 18, 35, 51, 56, and 59. Three novel non-synonymous variants, E148K in HPV35, S63C in HPV16 E7 and S495F in HPV18 L1, were identified, expanding the catalog of African HR-HPV diversity. Despite the limited sequence availability and snapshot nature of the data, our results highlight genetic diversity that may be relevant to vaccine design, though functional effects on vaccine efficacy remain to be established. Strengthening genomic surveillance and expanding sequencing efforts in Africa are critical to monitor HPV distribution, emerging variants, and to guide the development of effective vaccines. The South African Vaccine Innovation and Manufacturing Strategy (VIMS) project [59] provides a key opportunity for collaborative efforts to address these challenges.

## Figures and Tables

**Figure 1 ijms-26-11056-f001:**
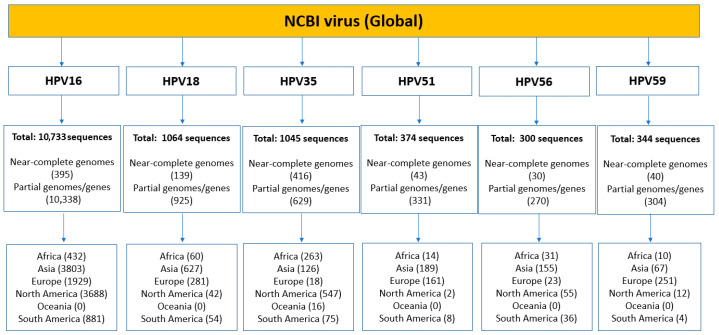
Pooled sequence data analysis of verified HR-HPV nucleotide sequences from the NCBI database. A breakdown of sequences for HPV types 16, 18, 35, 51, 56, and 59, focusing on those most prevalent in African populations. Data are categorized by geographic region, highlighting the distribution of sequences across continents, including near-complete (≥80%) genomes, partial genomes and genes.

**Figure 2 ijms-26-11056-f002:**
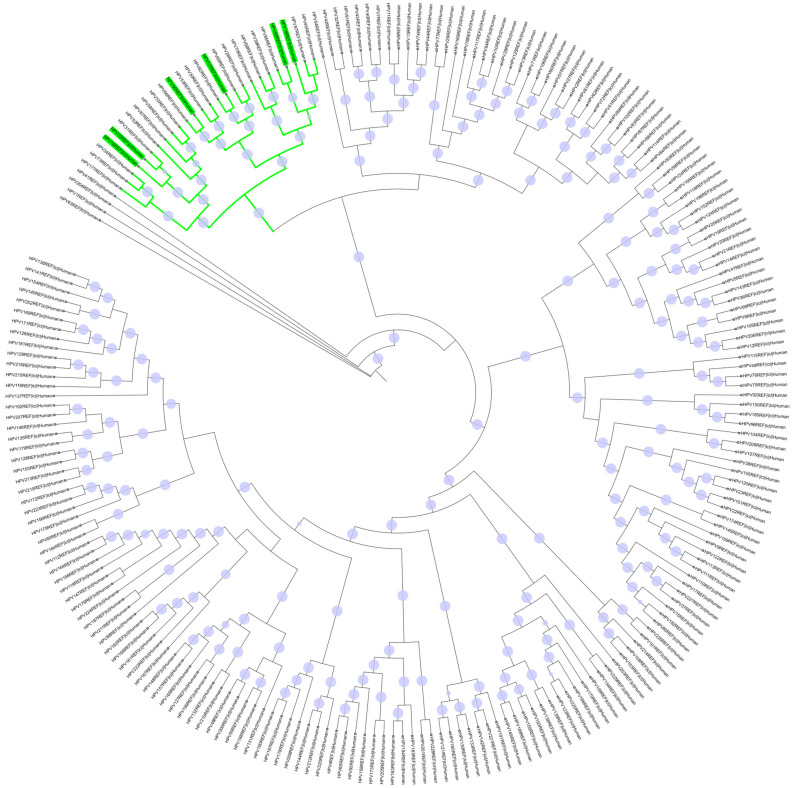
Maximum likelihood tree topology of 222 HR-HPV reference genomes. Multiple sequence alignments were performed using MAFFT v7.525, and phylogenetic analysis was performed using IQ-TREE v2.0.7 with SH-aLRT/UFboot tests for 1000 replicates. Purple dots indicate branches with support values ≥ 70%, where larger dots represent relatively higher support values. HPVs 16, 18, 35, 51, 56, and 59 are highlighted in green and cluster within the same monophyletic clade.

**Figure 3 ijms-26-11056-f003:**
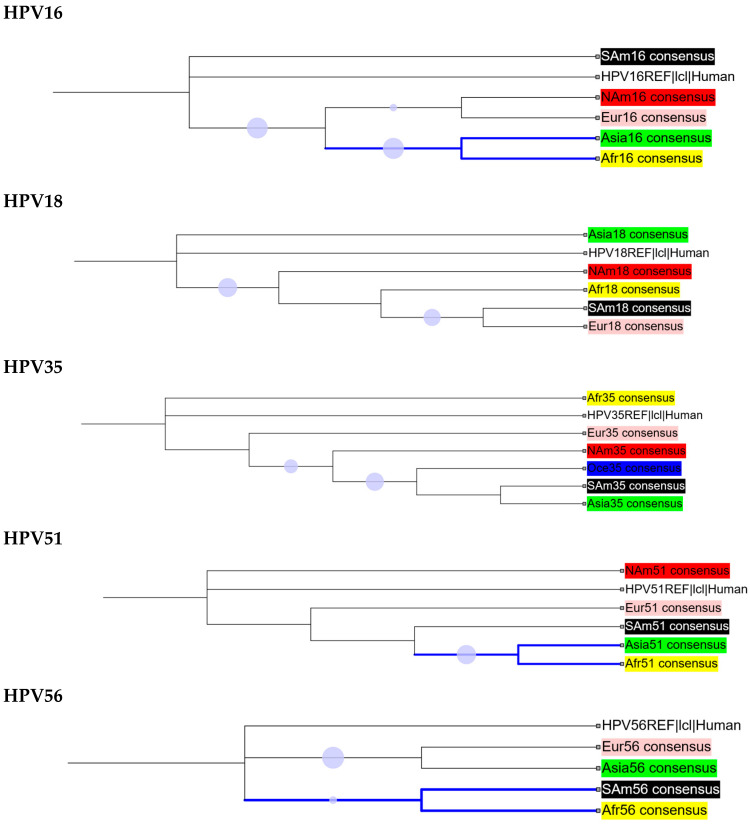
Maximum likelihood tree topologies of HR-HPV consensus sequences from different regions of the world. Multiple sequence alignments were performed using MAFFT v7.525, and phylogenetic analysis was conducted using IQ-TREE v2.0.7 with SH-aLRT/UFboot tests for 1000 replicates. Purple dots indicate branches with support values ≥70%, with larger dots representing relatively higher support values. Branches showing notable clustering are shown in blue.

**Figure 4 ijms-26-11056-f004:**
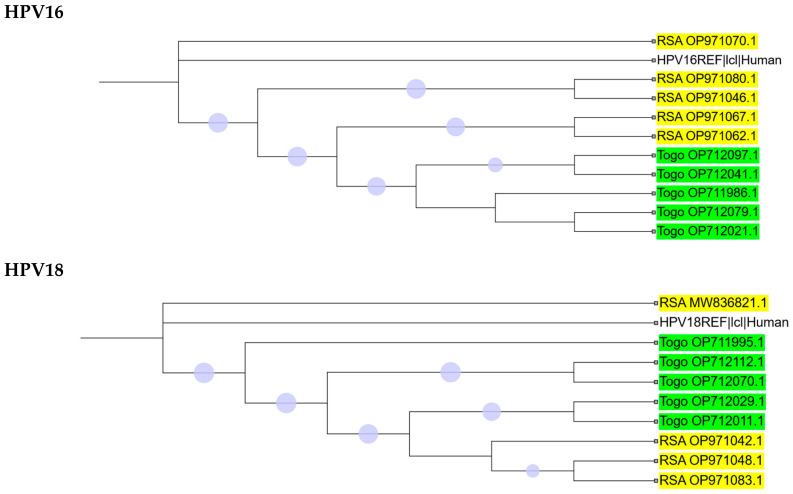
Maximum likelihood tree topologies of near-complete (≥80%) genomes of HR-HPV isolates from countries across Africa. Multiple sequence alignments were performed using MAFFT v7.525, followed by phylogenetic analysis using IQ-TREE v2.0.7 with SH-aLRT/UFboot tests for 1000 replicates. Purple dots represent branches supported by values ≥ 70%, with larger dots indicating relatively higher support. Branches showing notable clustering in HPV35 are shown in blue.

**Figure 5 ijms-26-11056-f005:**
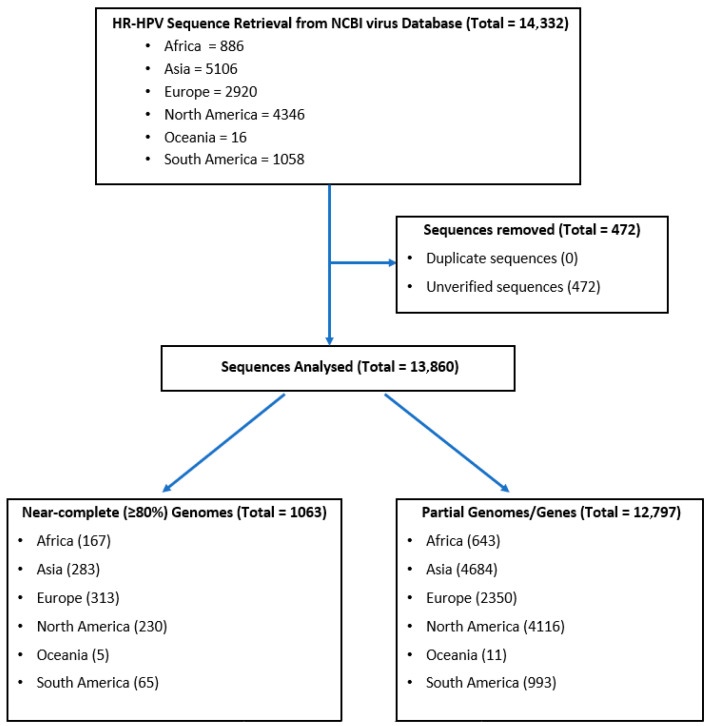
Flow diagram showing the data collection and sorting procedure for the study. Nucleotide sequences of six HR-HPV types (16, 18, 35, 51, 56, and 59) from different regions of the world were retrieved from the NCBI Virus database.

**Table 1 ijms-26-11056-t001:** Full-length E5, E6, E7, and L1 gene sequences of HR-HPV types in Africa retrieved from the NCBI Virus Database.

HPV	Gene Size (Bases) ^a^	Protein Length (Amino Acids)	Geographical Distribution	Number of Sequences (N)
**E5**				
**16**	252	83	South Africa (5), Togo (6)	11
**18**	222	73	South Africa (4), Togo (6)	10
**35**	252	83	Algeria (5), Guinea (27), Kenya (8), Mali (4), Morocco (2), Nigeria (28), Rwanda (92), South Africa (14), Togo (2), Zimbabwe (7)	189
**51**	NA	NA	NA	NA
**56**	NA	NA	NA	NA
**59**	222	73	Togo (10)	10
**TOTAL**				220
**E6**				
**16**	456	151	South Africa (5), Togo (6)	11
**18**	477	158	Nigeria (2), South Africa (4), Togo (5)	11
**35**	450	149	Algeria (5), Central Africa Republic (4), Chad (4), Guinea (24), Kenya (9), Mali (4), Morocco (2), Nigeria (25), Rwanda (82), South Africa (18), Togo (2), Zimbabwe (8)	187
**51**	456	151	South Africa (4), Togo (8)	12
**56**	468	155	South Africa (4), Togo (7)	11
**59**	483	160	Togo (10)	10
**TOTAL**				242
**E7**				
**16**	297	98	Republic of the Congo (2), South Africa (5), Togo (6)	13
**18**	318	105	South Africa (4), Togo (5)	9
**35**	300	99	Algeria (5), Guinea (26), Kenya (10), Mali (4), Morocco (2), Nigeria (27), Rwanda (94), South Africa (18), Togo (2), Zimbabwe (8)	196
**51**	306	101	South Africa (4), Togo (8)	12
**56**	318	105	South Africa (4), Togo (7)	11
**59**	324	107	Togo (10)	10
**TOTAL**				251
**L1**				
**16**	1518	505	South Africa (6), Togo (7)	13
**18**	1524	507	South Africa (4), Togo (7)	11
**35**	1509	502	Algeria (6), Guinea (27), Kenya (10), Mali (4), Morocco (2), Nigeria (28), Rwanda (86), South Africa (18), Togo (2), Zimbabwe (8)	191
**51**	1515	504	South Africa (4), Togo (9)	13
**56**	1500	499	South Africa (4), Togo (7)	11
**59**	1527	508	Togo (10)	10
**TOTAL**				249

^a^: Gene length includes start and stop codons; NA: Not applicable.

**Table 2 ijms-26-11056-t002:** Summary of E5 Protein Amino Acid Polymorphisms Identified in Africa. Synonymous and non-synonymous SNPs were detected in the E5 gene sequences of HR-HPV types from African populations using the variant-calling tool in Geneious Prime^®^.

HPV	Number of Variants	
	Synonymous	Non-Synonymous
*16*	6	5
*18*	2	4
*35*	6	9
*51*	N/A	N/A
*56*	N/A	N/A
*59*	2	5
*Total*	16	23

**Table 3 ijms-26-11056-t003:** Summary of E6 Protein Amino Acid Polymorphisms Identified in Africa. Synonymous and non-synonymous SNPs were detected in the E6 gene sequences of HR-HPV types from African populations using the variant-calling tool in Geneious Prime^®^.

HPV	Number of Variants	
	Synonymous	Non-Synonymous
*16*	4	6
*18*	7	1
*35*	9	10
*51*	3	2
*56*	4	3
*59*	4	1
*Total*	31	23

**Table 4 ijms-26-11056-t004:** Summary of E7 Protein Amino Acid Polymorphisms Identified in Africa. Synonymous and non-synonymous SNPs were detected in the E7 gene sequences of HR-HPV types from African populations using the variant-calling tool in Geneious Prime^®^.

HPV	Number of Variants	
	Synonymous	Non-Synonymous
*16*	3	2
*18*	2	3
*35*	3	4
*51*	1	3
*56*	0	5
*59*	1	4
*Total*	10	21

**Table 5 ijms-26-11056-t005:** Summary of L1 Protein Amino Acid Polymorphisms Identified in Africa. Synonymous and non-synonymous SNPs were detected in the L1 gene sequences of HR-HPV types from African populations using the variant-calling tool in Geneious Prime^®^.

HPV	Number of Variants	
	Synonymous	Non-Synonymous
*16*	17	9
*18*	15	5
*35*	44	10
*51*	25	7
*56*	13	4
*59*	15	19
*Total*	129	54

## Data Availability

The datasets analyzed during the current study are available in the National Center for Biotechnology Information (NCBI) repository, https://www.ncbi.nlm.nih.gov/. Accession numbers of sequences analyzed are provided in the Appendix A.

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
