# Peer review of "Genetic Diversity of Selected High-Risk HPV Types Prevalent in Africa and Not Covered by Current Vaccines: A Pooled Sequence Data Analysis"

_ijms, 2025, doi:10.3390/ijms262211056_

Round 1

Reviewer 1 Report

Comments and Suggestions for Authors

Comments on abstract

The abstract provides a comprehensive overview of the study and covers the key aspects of background, methods, results, and implications. However, there are several points that need attention:

  1. Irrelevant detail: It is unnecessary to indicate the exact word count (“191 words”) at the beginning of the abstract. This information is irrelevant to the scientific content and should be removed.
  2. Clarity and conciseness: While informative, the abstract could be streamlined to improve readability. For example, some sentences (e.g., the one beginning with “Furthermore, the genetic diversity of HR-HPV oncoproteins…”) could be shortened without losing meaning.
  3. Methods description: The mention of specific software and versions (e.g., Geneious Prime® 2024.0.7) is too technical for an abstract. Such details should be kept in the Methods section.
  4. Results emphasis: The identification of evolutionary relationships (HPV16 and HPV35) and novel variants (e.g., E148K in HPV35 E6, S495F in HPV18 L1) is valuable. These contributions should be framed more clearly as novel and significant findings.
  5. Conclusions and implications: The final statement about the need for Africa-specific vaccines is highly relevant but could be strengthened by explicitly linking the findings to gaps in current vaccine design and public health priorities.
  6. Keywords: The current set is adequate, though the addition of terms such as “genetic variability” or “phylogenetics” would improve searchability.

Summary: Overall, the abstract is strong but should remove irrelevant details (word count, software versions), be slightly shortened, and more explicitly highlight the novelty and public health importance of the findings.

Comments on the Introduction

The introduction provides a solid background on the global and regional burden of HPV and cervical cancer. It contextualizes the prevalence of different high-risk HPV types worldwide and in Africa, and correctly identifies the gap in vaccine coverage for types 35, 51, 56, and 59. However, several aspects could be improved:

  1. Focus and conciseness: The introduction is informative but could be more concise. Some sentences are overloaded with epidemiological percentages and statistics that may distract from the main narrative. Consider summarizing some of these details and highlighting only the most relevant contrasts between global and African prevalence.
  2. Redundancy: The introduction repeats the global versus African prevalence patterns in several parts. This could be streamlined to avoid redundancy and maintain reader engagement.
  3. Clarity of objectives: The introduction sets the stage for the study but the transition to the research objective could be more explicit. The final paragraph does state the aim (characterizing genetic diversity of E5, E6, E7, and L1 in African HPV types), but the rationale for why these proteins are specifically important (especially L1 versus the oncoproteins) could be explained more clearly.
  4. References: The section is well supported by references, but some citations appear outdated (e.g., Stanley et al., 2007) when more recent reviews exist. Incorporating the latest literature would strengthen the justification.
  5. Framing of significance: The introduction emphasizes the lack of vaccine coverage for African-relevant HPV types, which is important. However, it would benefit from explicitly stating how this study fills a gap in knowledge compared to previous “isolated studies” and systematic analyses.

Comments on Methods and Materials

  1. Inconsistent software versions (reproducibility risk).
    You report using Geneious Prime® 2024.0.7 for sequence handling/annotation, but Clustal Omega runs are said to have been executed in Geneious Prime® 2025.0.2. Please reconcile and state precisely which steps ran on which version, or standardize to a single version for the whole pipeline.
  2. Ambiguous/incorrect count of reference genomes retrieved.
    The text says “Complete reference genomes for each of the four HR-HPV types were retrieved,” but the study targets six types (16, 18, 35, 51, 56, 59). Correct this and confirm that reference sequences for all six were used.
  3. Phylogenetic methodology is underpowered and insufficiently justified.
    • Using neighbor-joining (NJ) with the Tamura–Nei model and only 100 bootstraps with a 50% support threshold is weak for whole-genome phylogenetics. Please (a) justify NJ over likelihood-based methods; (b) increase replicates (≥1000) and raise the reporting threshold (≥70%); and (c) report model selection (e.g., ModelFinder) or cite the original TN reference rather than an unrelated 2025 paper. Consider rerunning with ML (IQ-TREE2/RAxML-NG) and SH-aLRT/UFboot supports.
  4. Multiple sequence alignment details are missing.
    You state Clustal Omega 1.2.2 was used but do not specify parameters (gap penalties, iteration, trimming of ambiguously aligned regions, masking strategies). For viral genomes (HPV), MAFFT-L-INS-i is commonly preferred. At minimum: report exact parameters and whether you trimmed poorly aligned sites before tree inference.
  5. Variant calling methodology is opaque (P-values unexplained).
    You “recorded type, frequency, and P-values” using Geneious’ “find variation/SNPs” feature, but do not define the statistical test, the null hypothesis, how sequences (not reads) map to independent observations, nor any multiple-testing correction (FDR). Please fully specify the test, independence assumptions (e.g., one sequence per subject), and correct for multiple comparisons. Otherwise, the reported P-values (many extremely small) are not interpretable.
  6. Undefined inclusion/exclusion and deduplication policy.
    You removed sequences with unverifiable origin, but do not define how “origin was verified” (GenBank country field? publication metadata?), nor how you handled duplicate submissions, lab clones, serial samples from the same subject, or near-identical sequences that could inflate frequencies. Please add explicit criteria and a deduplication strategy.
  7. Time-window and snapshot bias.
    Retrieval was limited to August–October 2024, which is fine, but you should acknowledge and mitigate snapshot bias (e.g., re-query date, completeness checks by region/type) and provide a PRISMA-style flow diagram of sequence counts through filtering.
  8. Section numbering/structure is confusing.
    The Methods subsections re-start numbering (“1.” appears multiple times; later you jump to “4. Variation Calling…”). Please renumber sequentially (1, 2, 3, 4) for clarity and reproducibility.
  9. Gene annotation and presence/absence criteria are not described.
    You manually annotated reference genomes in Geneious and state E5 is absent for HPV51/56 (consistent with references), but you should still describe how ORF presence/absence was verified across isolates (e.g., ORF length thresholds, start/stop codons, frameshifts), and whether any automated checks (Prokka/ORFfinder) corroborated manual annotation.
  10. Definition of “full-length” and quality thresholds.
    You exclude sequences “without identifiable start or stop codons,” but do not define minimal coverage/length, %N thresholds, or ambiguity filters. Please add explicit QC thresholds for each gene (E5/E6/E7/L1) and for complete genomes.
  11. Terminology: “sequence-level meta-analysis.”
    The work is a pooled database analysis rather than a formal meta-analysis of studies. Consider revising wording or add a justification/definition for “meta-analysis” at the sequence level, including how you handled study-level heterogeneity (you did not).
  12. Reproducibility: accession lists and code/workflow.
    Methods should link a complete list of accession numbers used (per figure/table) and provide the analysis workflow (scripts, settings files, random seeds) in a repository. Currently, this is missing; iTOL visualization alone is not reproducible.
  13. Denominator inconsistencies (spillover from Results).
    While in Results, the L1 counts contradict themselves (249 full-length L1 sequences vs “962 full-length L1 gene sequences analyzed”). Please ensure Methods define denominators clearly and that Results report consistent totals derived from those definitions.
  14. Geographic coding is manual—add validation.
    You assigned region/country codes by manual renaming. Please state how you validated against metadata (and by whom/how many reviewers), and whether any automated checks ensured consistency.
  15. Model references.
    The Tamura–Nei model citation (“Duong et al., 2025”) is not the appropriate methodological source. Replace with the original model reference and/or a phylogenetics methods reference; keep the influenza paper, if relevant, in Discussion, not as a methods authority.
  16. Alignment scope and masking for phylogeny.
    You state that phylogeny used “all verified complete genomes… with respective reference genomes.” Please clarify whether coding and non-coding regions were included, how you handled indels and LCR, and whether you masked hypervariable or recombinant regions before tree building.

Comments on the Results

  1. Inconsistent sequence counts (critical).

    • The text reports retrieval of 14,332 sequences in total, of which 1,036 were complete genomes and 13,299 partial genomes/genes. Later, the L1 dataset is described as 249 full-length L1 sequences, but also as 962 full-length L1 sequences in the same section. This internal inconsistency undermines trust in the denominators used for frequency calculations. Please correct and clearly reconcile these counts.

  2. Overrepresentation/underrepresentation not addressed.
    HPV16, HPV18, and HPV35 are dominant in the datasets, while HPV51, HPV56, and HPV59 are extremely underrepresented. Yet, results are presented with equal narrative weight. Please provide normalization or at least cautionary notes about the interpretability of under-sampled types (e.g., HPV59 with only 40 sequences).

  3. Tables (gene polymorphism summaries) are unreadable in current form.

    • Tables 2–5 list dozens of variants with positions, amino acid changes, frequencies, and P-values, but the formatting is overwhelming and impossible to interpret without careful cross-referencing.

    • Many variants are reported at <1% frequency (singletons) yet still assigned P-values, which makes little statistical sense given the lack of replicate reads. These should be clearly separated as rare/singleton observations.

    • Suggest condensing the tables into (a) summary tables with counts of synonymous vs non-synonymous variants per gene per type, and (b) a supplementary file with the exhaustive variant lists.

  4. Use of P-values is misleading.
    The Results section repeatedly interprets variants with P-values as low as 10^-22 (e.g., Table 2, Table 3). Since these were computed on static consensus sequences rather than sequencing reads, the statistical meaning of these values is unclear. Reporting them gives a false sense of rigor and should be avoided unless the statistical test is properly justified.

  5. Figures — readability issues (serious).

    • Figure 1: The color-coded bars are too small to distinguish, especially for regions with few sequences. The figure legend does not provide numerical values, forcing the reader to guess proportions from tiny slivers.

    • Figures 2–4 (phylogenetic trees):

      • The resolution in the current PDF is poor; individual isolate labels, bootstrap values, and color codes are illegible.

      • In some trees, the branches are too compressed, and the bootstrap support values (threshold 50%) are not even visible. Many collapsed nodes make interpretation very difficult.

      • Without clear visualization, the claim of “close evolutionary relationship between HPV16 and HPV35” cannot be critically evaluated.

      • Recommendation: deposit high-resolution, zoomable trees (SVG or PDF with vector graphics) as supplementary files, and in the main manuscript provide simplified summary trees highlighting the key relationships.

  6. Figure 3 and 4 (regional clustering claims) not fully supported.

    • You state that African isolates cluster with European, Asian, or South American isolates, but the figures are too dense and poorly labeled to verify.

    • Nodes with <70% bootstrap support should not be interpreted, yet your text interprets several.

  7. Variant reporting lacks structure.

    • The Results section moves abruptly from “phylogenetic tree topologies” to “E5/E6/E7/L1 variants” without a clear structure. It would improve readability to divide results into (a) sequence dataset composition, (b) phylogenetic analyses, (c) intra-African diversity, (d) oncoprotein/L1 variant analysis.

    • As written, the section feels like a “data dump” rather than a structured narrative.

  8. Overclaiming evolutionary insights.
    The Results state that HPV16 and HPV35 “exhibit close evolutionary relationships, indicating shared traits that could impact vaccine efficacy.” This is speculative and belongs in the Discussion, not the Results. The tree shows clustering, but the functional implications cannot be inferred directly from phylogeny.

  9. Figures not linked to results text.
    Many statements (“HPV59 isolates clustered with North American isolates” etc.) are not annotated in the figure with arrows or highlights, forcing readers to guess. Please annotate trees directly to show which statements are supported.

  10. Underreporting of metadata (critical missing context).
    The Results present country-level clustering, but no temporal metadata are given (years of collection). Without temporal context, claims of diversity and evolution are incomplete. At minimum, a table of sequence collection years per type/country should be presented.

Comments on the Discussion

  1. Overly speculative claims.

    • The Discussion repeatedly infers functional or vaccine-related consequences directly from phylogenetic clustering (e.g., “HPV35 may share infectivity traits with HPV16, potentially enabling it to evade vaccine protection”). Such statements are speculative without experimental data. Please reframe them as hypotheses, not conclusions.

  2. Unsupported evolutionary narrative.

    • The invocation of Darwin’s natural selection (Bateson, 2017) to explain HPV strain dominance is simplistic and unnecessary. A more precise evolutionary framework (e.g., molecular epidemiology, selection pressure analysis on E6/E7) should be used, or the section should be cut to avoid overgeneralization.

  3. Overinterpretation of limited data.

    • HPV35 is highlighted as “becoming more prevalent in Africa,” yet the dataset is derived from a non-random sample of GenBank sequences (biased by availability, lab focus, and study designs). Without incidence data, prevalence trends cannot be inferred. Please temper this claim and explicitly acknowledge database bias.

  4. Misuse of P-values.

    • The Discussion interprets ultra-low P-values (10^-22) as evidence of important polymorphisms (e.g., T266A in L1, Q14H in E6). As noted in Results, these values are not meaningful when applied to consensus sequences. Please remove statistical interpretation and instead report variant frequencies descriptively.

  5. Redundancy and repetition.

    • Many points repeat earlier Results (e.g., HPV16/35 clustering, variants in E6/E7). The Discussion should focus on interpretation, implications, and comparison to literature, not rehash detailed variant lists. Consider condensing.

  6. Insufficient consideration of limitations.

    • The main limitations are mentioned but downplayed. They must be expanded:

      • Sampling bias: sequences in GenBank are heavily skewed toward a few regions (e.g., South Africa, Togo).

      • Underrepresentation: HPV51, HPV56, HPV59 data are insufficient for robust conclusions.

      • Lack of clinical metadata: No lesion grade, patient age, or temporal data were analyzed, limiting epidemiological relevance.

      • Reliance on in silico analysis: No functional validation of mutations.

    • These should be clearly acknowledged before making vaccine recommendations.

  7. Misplaced citations.

    • Several references are not appropriate for the claims (e.g., Duong et al., 2025, about avian influenza, cited in Methods/Discussion). Please ensure all citations directly support the argument made.

  8. L1 variation and vaccine implications overstated.

    • The claim that L1 polymorphisms (e.g., T266A, S282P) “may challenge vaccine efficacy” is overstated. Structural modeling or neutralization assays would be required to make this claim. Please soften the language to: “may warrant further evaluation regarding their potential impact on antigenicity.”

  9. Absence of comparison to other regions.

    • The Discussion notes that African HPV variants are diverse, but does not thoroughly compare their frequency spectrum to other continents. A more systematic comparative analysis (with percentages across regions) would strengthen the argument.

Comments on the Conclusion

  1. Too broad given the data.
    • The Conclusion states that findings “raise concerns for the long-term effectiveness of current vaccines” and emphasizes the need for “Africa-specific vaccines.” While this is a valid public health concern, the present dataset is not sufficient to support such a strong policy recommendation. It should be framed more cautiously.
  2. Restates rather than synthesizes.
    • The Conclusion largely repeats points from the Abstract and Discussion without synthesizing how this study advances knowledge. It should highlight:
      • Novelty (e.g., reporting of rare variants like E148K in HPV35 E6, S495F in HPV18 L1).
      • Limitations (biased dataset, no functional validation).
      • Next steps (functional characterization, improved genomic surveillance, broader African sampling).
  3. Missed opportunity to contextualize impact.
    • The section should explicitly address how these results could guide future genomic surveillance, inform public health agencies, or refine vaccine candidate prioritization. Currently, it is too general.
  4. Ambiguity in scope.
    • Statements such as “investing in genomic surveillance will be critical” are correct but vague. It would be stronger to propose concrete steps (e.g., establishment of regional HPV sequencing consortia, integration with WHO HPV surveillance frameworks).
Comments on the Quality of English Language

The manuscript is generally understandable, but the quality of English requires significant improvement before publication. The main issues include:

  1. Grammar and syntax: Several sentences are overly long, contain misplaced modifiers, or have awkward constructions that reduce clarity. Shorter, more direct sentences would improve readability.

  2. Repetition: The same ideas (e.g., HPV16 and HPV35 evolutionary relationship, prevalence differences between Africa and global estimates) are repeated multiple times across Abstract, Introduction, Results, and Discussion. This redundancy should be eliminated.

  3. Word choice and tone: Some phrases are colloquial or speculative (e.g., invoking Darwin’s natural selection in a simplistic way) and should be replaced with precise scientific wording.

  4. Terminology consistency: Terms such as “sequence-level meta-analysis,” “complete genomes,” “variants,” and “mutations” are used interchangeably, sometimes inaccurately. Consistent terminology is needed.

  5. Formatting and clarity:

    • The use of abbreviations (HR-HPV, L1, E6/E7, etc.) is correct but should be defined only once and then applied consistently.

    • Section numbering in Methods is inconsistent and confusing.

    • Tables and figures require better captions with clear explanations.

Author Response

Reviewer Comment

Author Response / Actions Taken

Abstract – Irrelevant detail: exact word count indicated

Removed word count from abstract.

Abstract – Clarity and conciseness: long sentences

Abstract revised; long sentence starting with “Furthermore, the genetic diversity of HR-HPV oncoproteins…” now reads: “The genetic diversity of HR-HPV oncoproteins in Africa remains poorly characterized, despite their potential as alternative vaccine targets.”

Abstract – Methods description too technical

Mentions of software versions removed; retained only in Methods section.

Abstract – Results emphasis: novelty of mutations

Results section revised to highlight novel non-synonymous mutations including E148K in HPV35 E6, S63C in HPV16 E7, S495F in HPV18 L1, and known oncogenic variants L83V (E6) and N29S (E7) in HPV16.

Abstract – Conclusions and implications

Conclusion revised to highlight the need for Africa-specific vaccines and genomic surveillance.

Abstract – Keywords

Replaced “diversity” and “cervical cancer” with “genetic variability” and “phylogenetics”.

Introduction – Focus and conciseness

Epidemiological statistics summarized; key points highlighted.

Introduction – Redundancy

Streamlined repetitive prevalence statements.

Introduction – Clarity of objectives

Explicitly described importance of E5, E6, E7, and L1 proteins and clarified the rationale and objectives.

Introduction – References

Updated outdated references with more recent literature.

Introduction – Significance

Added statement on lack of systematic analysis and study objective.

Methods – Software version inconsistencies

Corrected to Geneious Prime v2025.0.7 for all steps.

Methods – Reference genome retrieval

Corrected to include all six HR-HPV types (16, 18, 35, 51, 56, 59).

Methods – Phylogenetic methodology

Phylogenetic tress redone and replaced NJ with ML trees using IQ-TREE2; 1000 replicates, UFboot and SH-aLRT supports.

Methods – Multiple sequence alignment

Multiple alignment redone and replaced Clustal Omega with Mafft v7.525 (FFT-NS-2, default settings); trimmed alignments using AliView.

Methods – Variant calling

The software calculated P-values are removed; variant frequencies are now reported with 95% CI; two-tailed Fisher’s exact test for geographic differences.

Methods – Inclusion/exclusion & deduplication

Excluded “Unverified” sequences; verified geographical origin; duplicates checked by accession numbers.

Methods – Snapshot bias

Limitation acknowledged; PRISMA-style flow diagram included.

Methods – Section numbering

Sequential numbering applied.

Methods – Gene annotation

Verified ORFs via GenBank annotations; manual checks with Geneious Prime “Find ORFs”.

Methods – Definition of “full-length”

Full-length genes required identifiable start/stop codons; coverage ≥100% for genes and ≥80% for complete genomes.

Methods – Terminology

“Sequence-level meta-analysis” replaced with “pooled sequence data analysis”.

Methods – Reproducibility

Accession numbers listed; workflow described; GUI-based analysis explained.

Results – Sequence count inconsistencies

Sequence count was corrected and ensured consistence; removed misleading word “L1”.

Results – Overrepresentation/underrepresentation

Added caution regarding under-represented types (HPV51, HPV56, HPV59).

Results – Tables readability

The tables in the manuscript were summarized; detailed tables with variants, variant frequencies, geographic location and re-calculated P-values have been moved to supplementary material.

Results – Figures

The quality of the figures has improved. We now have High-resolution, zoomable trees. We have summary trees (consensus sequences per region in the world and consensus sequences from countries in Africa) in main manuscript; detailed trees of individual sequences have been moved to supplementary material. The legends have also improved.

Results – Evolutionary claims

Speculative claims removed; interpretations labeled as hypotheses.

Results – Metadata

Temporal limitations acknowledged; ML trees interpreted for relationships.

Discussion – Overly speculative claims

Reframed as hypotheses using “may” and “could”.

Discussion – Unsupported evolutionary narrative

Darwin reference removed; focus on molecular epidemiology.

Discussion – Overinterpretation

Database bias acknowledged; removed statements about prevalence trends.

Discussion – Misuse of P-values

Removed; reporting based on 95% CI.

Discussion – Redundancy

Variant discussion condensed; only relevant variants highlighted with literature context.

Discussion – Limitations

We have expanded on this and acknowledged the study limitations.

Discussion – Vaccine implications

Statements softened: “may warrant further evaluation regarding potential impact on antigenicity.”

Discussion – Comparison to other regions

Additional studies cited for context; focus on Africa maintained; follow-up studies planned.

Conclusion – Overbroad claims

Statements on vaccine effectiveness tempered; focus on implications for surveillance and research.

Conclusion – Synthesis & impact

Highlighted novelty, limitations, next steps, and practical implications.

Quality of English

Reviewed and corrected grammar, syntax, word choice, tone; abbreviations standardized; redundancies removed; formatting improved. Quality of the figures and tables have also improved.

Reviewer 2 Report

Comments and Suggestions for Authors

Review of the manuscript

Cervical cancer represents a significant global health burden and is strongly associated with high-risk human papillomavirus (HPV) infection. In addition to cervical cancer, high-risk HPV infections are also implicated in other, less common cancers such as anogenital, head and neck, and penile cancers. The careful design of effective HPV vaccination strategies remains one of the most promising preventive measures against these malignancies.

Globally, the most prevalent HPV types include HPV 16 and 18, but also types such as 31, 33, 35, 39, among others. Studies focusing on specific geographical regions are essential, as they provide more accurate insights into the distribution of HPV subtypes and their clinical relevance.

This manuscript presents detailed research on HPV prevalence across several African countries—a region often underrepresented in global analyses due to economic, sociological, and cultural factors. The study offers valuable findings based on a wide range of analyzed samples. It also contributes important phylogenetic analyses of different HPV types, including sequence variations in viral oncogenes (E5, E6, and E7) and the capsid protein L1. Particularly noteworthy are the conclusions regarding the clinical significance of HPV types 35, 51, 56, and 59 in Africa, which differs from global trends.

While short-term analyses focusing mainly on vaccination against HPV 16 and 18 yield satisfactory results, the long-term perspective—evaluating future cancer incidence—emphasizes the necessity of expanding preventive strategies to cover other high-risk HPV subtypes relevant to the African population.

Recommended improvements to the manuscript:

  1. Introduction
    • Please add information about current vaccines and new vaccine candidates under development.
  2. Methods and Materials
    • The numbering of subsections is inconsistent (e.g., 1. Database Mining and Nucleotide Sequence Retrieval; 1. Sequence Data Sorting and Annotation). Please correct this issue.
  3. Results presentation
    • Subsection numbering again repeats (e.g., 1. Individual Sequence Level Meta-Analysis; 1. The global genetic diversity of HR-HPVs). Please revise.
    • Figure 1 (Meta-analysis of HR-HPV nucleotide sequences from the NCBI database) is incomplete.
    • Figure 2 (Phylogenetic tree topology of 222 HPV reference genomes) is very difficult to read, as the description around the circumference is illegible. The same issue applies to Figure 3, where the circles and their descriptions remain blurry even at higher magnifications, and the figure legends are incomplete.
    • Consider alternative methods of results presentation (e.g., supplementary materials), as the current figures and tables are too large and complex to be presented clearly in the main text.
  4. Conclusion
    • Please provide a clear statement on which specific parameters and HPV variants, in the authors’ opinion, should be prioritized for inclusion in vaccine development strategies tailored specifically for Africa.

Author Response

Section

Reviewer Comment

Author Reply

Introduction

Please add information about current vaccines and new vaccine candidates under development.

Paragraph added describing existing HPV vaccines and new vaccine development efforts.

Methods and Materials

Inconsistent subsection numbering.

Numbering corrected and standardized.

Results Presentation

Repeated subsection numbering.

Corrected for sequential consistency.

Figure 1 incomplete.

Figure resized to fit manuscript.

Figures 2 and 3 difficult to read; legends incomplete.

High-resolution images provided both in the manuscript and as supplementary material; legends expanded.

Consider supplementary materials for large tables/figures.

Detailed variant tables moved to Supplementary Material; summary tables retained in main text.

Detailed figure moved to Supplementary; summary figures retained in main text

Discussion/Conclusion

Specify which HPV variants should be prioritized for Africa-specific vaccines.

Recommendation added to prioritize HPV types 35, 51, 56, and 59 in addition to HPV16 and 18.

Round 2

Reviewer 1 Report

Comments and Suggestions for Authors

General assessment.
Thank you for the revised version. The manuscript shows substantial improvements in analytical design (phylogeny with stronger support and clearer inclusion criteria), consistency of gene/type counts, and organization of results. The supplementary material adds valuable transparency. With minor adjustments on reproducibility, figure quality, and cautious wording, the study will be ready for publication.

Remaining suggestions (all minor and actionable).

  1. Complete reproducibility.

    • Add accession lists (CSV/TXT) for each analysis/figure/table, as well as the exact commands/parameters (alignments and trees) and, if applicable, the random seed.

    • Clarify the definition of “complete genome” vs. “near-complete”: either raise the coverage threshold to justify the term “complete” or consistently use “near-complete (≥X%).”

  2. Quality control and deduplication (Methods).

    • Specify QC thresholds: minimum length per gene, maximum %N/ambiguity, treatment of indels.

    • Describe the deduplication policy (e.g., removal of identical accessions from the same subject/laboratory) and the geocoding validation process (double check by two reviewers, cross-check with metadata).

  3. Alignments and masking.

    • Justify the decision not to mask hypervariable/LCR regions, or alternatively include a sensitivity analysis (with vs. without masking) to demonstrate topological stability.

    • Standardize the description of gap penalties and trimming rules (state explicit criteria).

  4. Figures (legibility).

    • Export trees in vector format (SVG/PDF); increase minimum font size; annotate key clades/relationships mentioned in the text directly in the figures (arrows/boxes) and restrict interpretation to nodes with ≥ declared support threshold.

    • In regional/country composition figures, add legends with totals to avoid visual estimation.

  5. Language and cautious interpretation.

    • Further soften speculative statements regarding potential vaccine escape inferred only from phylogenetic clustering. A suggested phrasing is provided below.

  6. Terminology and formatting.

    • Use “pooled database analysis” or similar consistently (avoid “sequence-level meta-analysis”).

    • Correct any remaining template placeholders and standardize software version citations.

    • When noting the absence of E5 in certain types, add the specific PaVE/reference source.

  7. Tables and rare variants.

    • Keep only summaries (synonymous vs. non-synonymous counts by gene/type) in the main text; move exhaustive variant lists to the supplement.

    • Clearly separate singletons/variants <1% from main analyses and avoid over-interpreting extremely low-frequency variants. If statistical tests are applied (e.g., Fisher for geographic distribution), indicate the test and multiple-testing correction or otherwise limit to descriptive reporting.

  8. Temporal context.

    • Provide a short table with collection years per type and region to contextualize diversity and potential temporal sampling biases.

Author Response

Remaining Suggestions (All Minor and Actionable)

  1. Complete Reproducibility

Reviewer comment:
Add accession lists (CSV/TXT) for each analysis/figure/table, as well as the exact commands/parameters (alignments and trees) and, if applicable, the random seed.

Response:
We have addressed this request by providing supplementary materials that include accession lists for all analyses, figures, and tables. A detailed record of all commands and parameters used for sequence alignments and phylogenetic tree construction, including the random seed where applicable, has also been included. Sequences with accession numbers used for variant calling are provided in the supplementary files, and accession numbers for sequences included in the phylogenetic trees are also displayed on the trees.

Reviewer comment:
Clarify the definition of “complete genome” vs. “near-complete”: either raise the coverage threshold to justify the term “complete” or consistently use “near-complete (≥X%).”

Response:
In this study, we now use the term ‘near-complete genomes’ to refer to sequences with ≥80% genome coverage. For consistency, the term ‘near-complete’ is used throughout the manuscript, including in Figure 2, rather than ‘complete genome,’ to avoid overstatement of coverage.

  1. Quality Control and Deduplication (Methods)

Reviewer comment:
Specify QC thresholds: minimum length per gene, maximum %N/ambiguity, treatment of indels.

Response:
We have clarified our quality control thresholds: only sequences with ≥80% reference coverage and ≤15% ambiguous bases (N) were included and insertions and deletions were treated as gaps in the alignments. This information has been added to the Methods section under ‘Nucleotide Sequence Alignments and Phylogenetic Analysis.’

Reviewer comment:
Describe the deduplication policy and geocoding validation.

Response:
We have added clarity on the removal of records with identical accession numbers. Under Database Mining and Nucleotide Sequence Retrieval, we now state:

“Microsoft Excel Professional Plus 2019 was used to check for duplicate entries. Duplicate entries, including identical accession numbers originating from the same subject or laboratory, were removed prior to downstream analyses. Geographic metadata for each sequence were validated through a two-step process: first, each location was independently verified by two authors; second, locations were cross-checked against the accompanying metadata to ensure consistency and accuracy. Only sequences with validated metadata were included in subsequent analyses.”

  1. Alignments and Masking

Reviewer comment:
Justify the decision not to mask hypervariable/LCR regions, or include a sensitivity analysis.

Response:

“Our study aimed to assess the genetic diversity of HR-HPVs in Africa at the whole-genome level, consistent with previously published methodologies (Liu et al., 2017). For this reason, we did not mask hypervariable regions, including the long control region (LCR), in the multiple sequence alignments. By including all genomic regions, we ensured that both conserved and variable sites contributed to the phylogenetic analyses. Subsequent analyses of individual genes were then performed to identify gene-specific variation, allowing us to capture both genome-wide and gene-level diversity.”

Reviewer comment:
Standardize the description of gap penalties and trimming rules.

Response:

“Multiple sequence alignments were performed using MAFFT v7.525 with the FFT-NS-2 algorithm, using a gap opening penalty of 1.53 and a gap extension penalty of 0.00. Insertions and deletions were treated as gaps in the alignments. After alignment, residues corresponding to gaps in the reference sequence were trimmed using AliView v1.30 to reduce poorly aligned regions, ensuring consistent treatment of gaps across all sequences.”

  1. Figures (Legibility)

Reviewer comment:
Export trees in vector format (SVG/PDF); increase minimum font size; annotate key clades/relationships; restrict interpretation to nodes with ≥ declared support threshold.

Response:

“All phylogenetic trees are included in SVG format, allowing zooming to view labels clearly. Increasing font size further would result in overlapping labels and reduced legibility. Interpretations are restricted to nodes with support values ≥70%, and branches corresponding to notable clustering are highlighted in blue to facilitate visualization of key clades and relationships.”

Reviewer comment:
In regional/country composition figures, add legends with totals to avoid visual estimation.

Response:
Total counts for each region and country per HR-HPV type have been added to the supplementary material (Table S1 and Figure S2).

  1. Language and Cautious Interpretation

Reviewer comment:
Soften speculative statements regarding potential vaccine escape inferred from phylogenetic clustering.

Response:
We reviewed the abstract, discussion, and conclusion, and revised to avoid overinterpretation:

  1. Abstract: Statements suggesting functional implications of phylogenetic relatedness (e.g., between HPV16 and HPV35) have been softened to indicate that observed evolutionary relationships may suggest shared traits relevant to vaccine design, while functional implications remain unvalidated.
  2. Discussion:
    • Sentences implying possible vaccine escape based solely on phylogenetic proximity have been revised. For example, “HPV35 could potentially evade existing vaccine protection” now reads:

“While this may indicate shared biological traits, any implications for vaccine protection are speculative and require experimental validation.”

    • Observations of L1 gene variation now clearly indicate that functional consequences for antigenicity or vaccine efficacy remain to be established. Phrases like “may have implications for antigenicity and vaccine efficacy” were revised to:

“Could potentially influence antigenicity, but further functional studies are needed to determine any impact on vaccine efficacy.”

  1. Conclusion: Language now highlights genetic diversity and novel variants without overextending interpretation to vaccine effectiveness. Statements such as “emphasize the potential implications for vaccine efficacy” were revised to:

“Highlight genetic diversity that may be relevant to vaccine design, though functional effects on vaccine efficacy remain to be established.”

  1. Terminology and Formatting

Reviewer comment:
Use “pooled database analysis” consistently; correct template placeholders; standardize software version citations; cite PaVE when noting absence of E5.

Response:

  1. Terminology: “Sequence-level meta-analysis” replaced with “pooled sequence data analysis” throughout.
  2. Formatting: All template placeholders corrected; software version citations standardized (MAFFT v7.525, IQ-TREE v2.0.7, AliView v1.30).
  3. E5 absence: Explicitly cited the Papillomavirus Episteme (PaVE) database for HPV51 and HPV56.

  1. Tables and Rare Variants

Reviewer comment:
Keep only summaries in the main text; move exhaustive variant lists to the supplement; clearly separate singletons/variants <1%.

Response:

  • Summary data (synonymous vs. non-synonymous variant counts by gene and HPV type) are in the main text. Detailed variant lists and rare mutations are in supplementary materials with cross-references.
  • Variants present at <1% frequency are reported descriptively and excluded from statistical testing. Two-tailed Fisher’s exact tests were applied only to variants with interpretable frequencies for geographic distribution. No multiple-testing correction was performed, and this limitation is explicitly stated.

  1. Temporal Context

Reviewer comment:
Provide a short table with collection years per type and region.

Response:
This table has been added to the supplementary material under the phylogenetic analysis section.

References

  • Liu, Y., Pan, Y., Gao, W., Ke, Y., & Lu, Z. (2017). Whole-genome analysis of human papillomavirus types 16, 18, and 58 isolated from cervical precancer and cancer samples in Chinese women. Scientific Reports, 7, 263.

Reviewer 2 Report

Comments and Suggestions for Authors

The authors of the paper have taken my comments into account. The results are presented in a clear and readable manner. The conclusions drawn demonstrate the scientific value of the work. I accept the work in its current form.

Author Response

Reviewer's Comment Comments and Suggestions for Authors

The authors of the paper have taken my comments into account. The results are presented in a clear and readable manner. The conclusions drawn demonstrate the scientific value of the work. I accept the work in its current form

Response

We sincerely thank the reviewer for their constructive feedback and positive evaluation of our work. We are grateful for the time and effort dedicated to reviewing our manuscript and are pleased that the revisions have addressed the concerns raised.